# Genetic drift, historic migration, and limited gene flow contributing to the subpopulation divergence in wild sea beet (*Beta vulgaris* ssp. *maritima* (L.) Arcang)

Muhammad Massub Tehseen[1], Nathan A. Wyatt[2], Melvin D. Bolton[2], Karen K. Fugate[2], Lisa S. Preister[2], Shengming Yang[3], Vanitharani Ramachandran[2], Xuehui Li[1], Chenggen Chu[2]*

1 Department of Plant Sciences, North Dakota State University, Fargo, ND, United States of America,
2 USDA-ARS, Edward T. Schafer Agricultural Research Center, Sugarbeet and Potato Research Unit, Fargo, ND, United States of America, 3 USDA-ARS, Edward T. Schafer Agricultural Research Center, Cereal Research Unit, Fargo, ND, United States of America

* chenggen.chu@usda.gov

**Data Availability Statement:** All relevant data are within the manuscript and its Supporting Information files. Raw data are publicly available

## Abstract

Cultivated beet (*Beta vulgaris* L. ssp. *vulgaris*) originated from sea beet (*B. vulgaris* ssp. *maritima* (L.) Arcang), a wild beet species widely distributed along the coasts of the Mediterranean Sea and Atlantic Ocean, as well as northern Africa. Understanding the evolution of sea beet will facilitate its efficient use in sugarbeet improvement. We used SNPs (single nucleotide polymorphisms) covering the whole genome to analyze 599 sea beet accessions collected from the north Atlantic Ocean and Mediterranean Sea coasts. All *B. maritima* accessions can be grouped into eight clusters with each corresponding to a specific geographic region. Clusters 2, 3 and 4 with accessions mainly collected from Mediterranean coasts are genetically close to each other as well as to Cluster 6 that contained mainly cultivated beet. Other clusters were relatively distinct from cultivated beets with Clusters 1 and 5 containing accessions from north Atlantic Ocean coasts, Clusters 7 and Cluster 8 mainly have accessions from northern Egypt and southern Europe, and northwest Morocco, respectively. Distribution of *B. maritima* subpopulations aligns well with the direction of marine currents that was considered a main dynamic force in spreading *B. maritima* during evolution. Estimation of genetic diversity indices supported the formation of *B. maritima* subpopulations due to local genetic drift, historic migration, and limited gene flow. Our results indicated that *B. maritima* originated from southern Europe and then spread to other regions through marine currents to form subpopulations. This research provides vital information for conserving, collecting, and utilizing wild sea beet to sustain sugarbeet improvement.

online through USDA-BeetBase (https://beetbase.
scinet.usda.gov/).

**Funding:** This research is funded by USDA-ARS
CRIS project No. 3060-21000-045-000D, the Beet
Sugar Development Foundation (BSDF), and the
Sugarbeet Research and Education Board of
Minnesota and North Dakota (SBREB). The funders
had no role in study design, data collection and
analysis, decision to publish, or preparation of the
manuscript.

**Competing interests:** The authors have declared
that no competing interests exist.

## Introduction

Domestication and continuous artificial selection in breeding efforts often leads to a reduction
in genetic diversity in modern crop cultivars [1–3]. The loss of genetic diversity has signifi-
cantly narrowed the genetic base of modern crop varieties [4], thereby reducing adaptability of
crops to cope with unfavourable conditions. To enhance the buffering capacity of crops against
environmental changes, it is crucial to broaden genetic diversity, thus facilitating the introduc-
tion of beneficial genetic variations. Sugarbeet (*Beta vulgaris* L. ssp. *vulgaris*) is an important
crop that provides a significant source of sucrose [5] but is facing a myriad of disease and pest
issues, and additionally is not well-suited for growth in diverse environments [6]. Low genetic
diversity in sugarbeet cultivars has become a critical issue that decreases sustainability of the
sugarbeet industry [7–9]. It is very important to mine new gene pools to bring in new genetic
variations for developing cultivars with more resilience to stresses.

Wild relatives of sugarbeet are likely important genetic sources for adaptive variations cru-
cial for the development of new cultivars [10,11]. Sea beet (*Beta vulgaris* ssp. *maritima* (L.)
Arcang., according to classification adopted by the US National Plant Germplasm System
(NPGS), referred to as *B. maritima* hereafter), the wild progenitor of cultivated beet, has
emerged as a primary reservoir of genetic diversity for the enhancement of sugarbeet [11–16].
To fully exploit the potential of *B. maritima* for sugarbeet improvement, a more profound
understanding of its ecological dynamics, geographical spread and ability to acclimate to the
environment is needed [17,18].

Wild *B. maritima* populations extend along the Mediterranean basin (Algeria, Egypt,
France, Greece, Israel, Italy, Lebanon, Morocco, Spain, Tunisia, Turkey, etc.) and the Atlantic
coast (Belgium, Denmark, France, Germany, Netherlands, Spain, United Kingdom, etc.), and
introduced to many countries in Asia, Central Europe, north and south America (https://
powo.science.kew.org/). The observed clinal pattern of *B. maritima* is likely a result of post-
glacial biogeographic processes along the Atlantic coast, with the Mediterranean Basin serving
as a population refuge during the last ice age [19]. Previous studies have reported a gradient of
variation during the evolution of *B. maritima*, including the shift from an annual to a biennial
growth habit [20–23]. In recent years, research has focused on identifying *B. maritima* acces-
sions to improve sugarbeet resistance to diseases such as Cercospora leaf spot caused by *Cer-
cospora beticola* Sacc. [24], rhizomania caused by beet necrotic yellow vein virus (BNYVV)
[25], sugar beet cyst nematode (*Heterodera schachtii* Schmidt) [26], powdery mildew caused
by the fungus *Erysiphe polygoni* (formerly *Erysiphe betae*) [27], Aphanomyces root rot caused
by *Aphanomyces cochlioides* [28], and yellow wilt caused by *Fusarium oxysporum* f. sp. *betae*
[29], the development of a smooth root phenotype [30], and salt and drought tolerance [31],
but utilization of the genetic diversity in *B. maritima* for sugarbeet improvement remains
limited.

Using SNPs (single nucleotide polymorphisms) from microarrays, Andrello et al. [32] ana-
lyzed 1,512 individuals from 1080 accessions of *Beta* germplasm which grouped all accessions
into nine clusters, revealing genetic structure in the germplasm partially corresponding to geo-
graphical patterns. However, due to confounding differences between sea beet and cultivated
beets, a single geographic origin for cultivated beet domestication could not be definitively
determined. High-throughput next-generation sequencing technologies offer opportunities
for the efficient discovery of SNPs that cover the entire genome to precisely calculate genetic
distance within wild germplasm collections or between wild germplasm and cultivated crops.
Tehseen et al. [16] used SNPs generated from a GBS (genotype-by-sequencing) platform to
analyze 1,928 accessions of *B. vulgaris* germplasm lines that included 607 *B. maritima* acces-
sions and found a subset of 329 *B. maritima* accessions that was more genetically diverse from

sugarbeet, indicating their potential for introducing novel genetic variations from *B. maritima* to enhance sugarbeet improvement. Sandell et al. [33] sequenced the genome of 606 *Beta* germplasm lines that included 239 *B. maritima* accessions and 285 sugarbeet lines and identified two genetically distinct groups of sea beet with one from the Atlantic coast and the other from the Mediterranean Sea area. Notably, accessions collected from Greece exhibited the closest genetic proximity to sugarbeet in that study. Therefore, genetic diversity studies provide an efficient way of identifying *B. maritima* accessions with greater potential for broadening the genetic base of sugarbeet.

Besides improving the efficient utilization of *B. maritima* accessions in sugarbeet breeding, genetic diversity analysis is expected to also reveal population structure and gene flow during evolution of sea beet, thereby providing information that is useful for conserving and maintaining genetic diversity in *B. maritima*. Preservation of *B. maritima* becomes urgent since the habitats of sea beet along seashores have been altered by the erection of barriers to preserve the seacoast and the recreational use of beaches and estuaries [34]. Since genetic diversity in wild germplasm changes over time due to mutation, selection, genetic drift, and gene flow [35], analysis of genetic resources regarding population distribution, population structure, allelic frequency change, genetic distance, origin of domestication, etc., all provide important information for conserving and maintaining diversity. Therefore, it is essential to use markers that covering the entire genome to accurately investigate diversity of all *B. maritima* accessions.

Using SNPs generated through a GBS platform, we evaluated all publicly available *B. maritima* accessions that were collected along the coasts of the Atlantic Ocean, extending latitudinal from the English Channel to the Moroccan coast next to the strait of Gibraltar and longitudinally across the Mediterranean basin from the Balearic to Levantine seas. The objectives in this study were: 1) to evaluate population structure of the worldwide *B. maritima* collection, 2) elucidate clinal variation and admixture patterns of *B. maritima* populations, 3) estimate dynamic gene flow during *B. maritima* evolution and sugarbeet domestication, and 4) generate knowledge that will assist in maintaining the genetic diversity of *B. maritima* to sustain sugarbeet improvement.

## Material and methods

### Plant materials

A total of 599 *B. maritima* accessions from the U.S. National Plant Germplasm System (NPGS) and USDA-ARS sugarbeet genetics program at Fargo, ND were used in the current research. Accessions were collected or introduced from 25 countries and were divided into seven regions of the world including Africa (58 accessions, native collected), Asia (six accessions, two accessions that were from China and Uzbekistan may be introduced), northern Europe (181 accessions, native collected), southern Europe (179 accessions, native collected), western Europe (153 accessions, native collected), and North America (22 accessions, may be introduced) (Tables 1 and S1). In addition, a set of 30 cultivated beet lines including three fodder beet, three table beet, and 24 sugarbeet that came from different origins (S1 Table) were used as a reference to compare genetic distance among *B. maritima* accessions. These 30 lines were selected according to phylogenetic analysis in Tehseen et al. [16] to represent cultivated beet.

### SNP genotyping through a genotype by sequencing (GBS) platform

All materials were grown in a greenhouse room at 18–27 ˚C under a 16-h light/8-h dark regime. For GBS analysis, 0.1 g of fresh leaves were collected after 4-leaf stage from 7 to 10 plants of each accession and immediately lyophilized in a Virtis Freezemobile 35EL (SP Scientific, Inc., Warminster, PA, USA) for two days. The dried tissue was ground to a fine powder

**Table 1. List and origin of 599 *B. maritima* accessions used in the current study with their putative geographic regions.**

| Region | Countries (number of lines) | Total |
|---|---|---|
| Africa | Egypt (26), Morocco (31), Tunisia (1) | 58 |
| Asia | China (1), Georgia (1), Uzbekistan (1), India (2), Israel (1) | 6 |
| Northern Europe | Denmark (21), Ireland (49), Jersey Island (2), UK (109) | 181 |
| Southern Europe | Croatia (1), Cyprus (1), Greece (56), Italy (100), Portugal (6), Spain (8), Turkey (7) | 179 |
| Western Europe | Belgium (3), France (145), Germany (2), Netherlands, (2), Guernsey Island (1) | 153 |
| North America | United States | 22 |

using a 1600 MiniG SPEX homogenizer (SPEX, Inc., Metuchen, NJ, USA). Genomic DNA was extracted from dried tissue using a KingFisher Flex DNA purification system (KingFisher, Inc., Falls Church, VA, USA). DNA samples were then fragmented using NsiI and BfaI restriction enzymes, and barcoded adapters were ligated to the DNA fragments to differentiate between each accession. GBS sequencing libraries were created using the method of Hilario et al. [36] by amplifying barcoded DNA using a 96-plex plate, then purifying and quantifying the PCR product before sequencing with an Illumina HiSeq 2000 system (Illumina, Inc., San Diego, CA, USA). The sequenced fragments were compared to the sugarbeet genome sequence assembly EL10.2 of sugarbeet line EL10 [37]. Raw SNP data was filtered to remove SNPs with > 20% missing data and genotype imputation was done using Beagle v5.0 [38], thereby reducing the missing data rate to 0% and retaining only bi-allelic SNPs.

## Preparing SNP data for population structure analysis

Management of high-throughput SNP data was carried out using the *wrapper* function of R package SambaR [39]. Briefly, raw data was imported as a.vcf file using the R package vcfR and then converted to a genlight object using the package adegenet v2.1.3 [40–42]. SNPs were filtered through function *filterdata* in R package SambaR to remove SNPs with a minor allele frequency less than 0.05. Linkage disequilibrium (LD) decay was calculated using PLINK v1.9 [43], and SNPs on each chromosome that were apart over the LD decay distance were selected through PLINK v1.9 for population structure analysis.

## Phylogenetic analysis of *B. maritima* accessions and cultivated beet lines

To create an overview of the genetic distance among *B. maritima* accessions and between wild sea beet and cultivated beet, Tassel v5.0 was used to generate a phylogenetic tree of all genotypes via UPGMA (unweighted pair group method with arithmetic mean) hierarchical clustering [44]. Genetic distances were calculated using the Interactive Tree Of Life (iTOL v5) with phylogenetic clade tree data [45].

## Population structure analysis

Population structure of 599 *B. maritima* accessions and 30 cultivated beet lines was determined using both the Bayesian-model based clustering method [46] and the non-model-based discriminant analysis of principal components (DAPC) [40]. The Bayesian-model based clustering was conducted using STRUCTURE v.2.3.4, with ten independent replicates run for each putative subpopulation with the number of subpopulations ($K$) set from 2 to 10 under the

admixture model. Burn-in period was set at 10,000 and MCMC (Markov chain Monte Carlo) replications were set at 50,000. Delta $K$ ($\Delta K$) was used to estimate the optimal number of subpopulations based on the change in log probability of data between successive structure iterations using Structure Harvester [47].

DAPC was determined using R package adegenet v2.1.3 [41], with the *k-means* function used to find the maximum number of subpopulations with maximizing variations and the optimum number of clusters (subpopulations) corresponding to the lowest BIC (Bayesian information criterion) value. The optimal number of clusters was used in the DAPC analysis via the *dapc* function, which transformed data using PCA (principal component analysis) and performed a discriminant analysis on the retained principal components to form linear combinations of SNP markers with the largest between-group variance and the smallest within-group variance, to describe the clusters, i.e. subpopulations. In both *k-means* and *dapc* analyses, 200 principal components were retained for sufficiently explaining 85% of the variance.

## Genetic diversity analyses among subpopulations

Genetic differences between subpopulations were estimated using the *stamppFst* function in R package StAMPP v.1.6.1 [48]. Genome-wide Fst was calculated according to Weir and Cockerham [49] to estimate the proportion of genetic variance in a subpopulation relative to the total genetic variance in the whole collection and to compare similarities among subpopulations. The function *stamppNeisD* in the same R package was utilized to measure Nei's genetic distance [50] among subpopulations, which is used to estimate effects of mutation and genetic drift accounting for allelic change during evolution.

To compare genomic variations among subpopulations, the average major allele frequency calculated within every 1-Mb region on each chromosome was compared among subpopulations. The variances of the average allele frequency within the same genomic region across subpopulations were used as indicators of the variation level among subpopulations. Average major allele frequency and their variance across subpopulations were calculated using Microsoft Excel. Genomic regions that have a variance of major allele frequency above the 75th percentile were considered as regions significantly varied across subpopulations during *B. maritima* divergence.

## Genetic diversity analyses within subpopulations

Within each subpopulation, nucleotide diversity ($\pi$) and Watterson estimator (allelic richness) were calculated using the R package SambaR [39] to measure genetic variation among individuals in the subpopulation and minor allele frequency. Tajima's D (balance between rare and common alleles) was calculated using the same tool to measure gene frequency deviation from neutral evolution, i,e. any selective pressure existing during *B. maritima* evolution. Since all sub-populations consisted of varying numbers of genotypes, scaled values of diversity indices were utilized to provide a more comprehensive explanation of the data.

## Results

### Genotypic markers distribution

GBS platform generated approximately 520,000 SNPs using EL10.2 as the reference genomic assembly. From these, a group of 148,317 SNP markers was selected by removing SNPs with minor allele frequency and greater than >20% missing data rate. Chromosomes 5 and 6 had the most markers with 19,115 (12.89%) and 19,140 (12.90%) markers, respectively, while chromosome 9 had the least with 14,277 (9.63%) SNPs. Across the entire genome, the marker

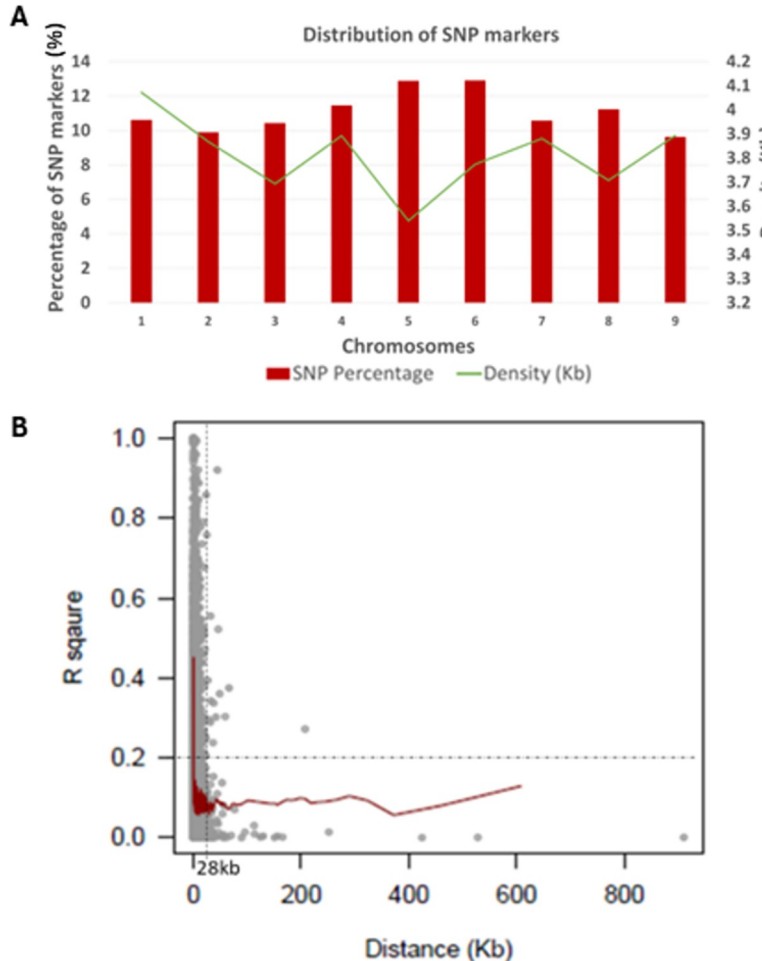

**Fig 1.** Distribution and density of single nucleotide polymorphisms (SNPs) across nine chromosomes (A) and average linkage disequilibrium (LD) decay (B) in the collection of 599 *Beta maritima* accessions and 30 cultivated beet lines.

density was, on average, 3.81 markers per kilobase (kb). Chromosome 5 had the highest marker density (3.54 markers/kb) while chromosome 1 had the lowest density (4.07 markers/kb) (Fig 1A). Analysis using the set of 148,317 SNPs in 599 *B. maritima* accessions and 30 cultivated beet lines found linkage disequilibrium (LD) decayed on average within 28 kb ($R^2 = 0.2$) (Fig 1B). A subset of 77,180 SNPs was obtained after LD pruning using threshold of $R^2 = 0.2$ and was then used for population structure analysis.

### Phylogenetic analysis of *B. maritima* accessions with cultivated beet lines

Phylogenetic analysis of 599 *B. maritima* accessions and 30 cultivated beet lines showed that *B. maritima* accessions were grouped into two distinct clusters, with one cluster distinct from cultivated lines and the other related to cultivated lines (Fig 2). The cluster related to cultivated lines contained 276 accessions with 60.9% of accessions collected from southern Europe, whereas the cultivated-distinct cluster included 323 accessions with 51.1% collected from northern Europe and 35.0% collected from western Europe (Table 2). This agreed with reports from Sandell et al. [33] and Tehseen et al. [16] that *B. maritima* accessions from northern and western Europe are more genetically diverse to cultivated beet.

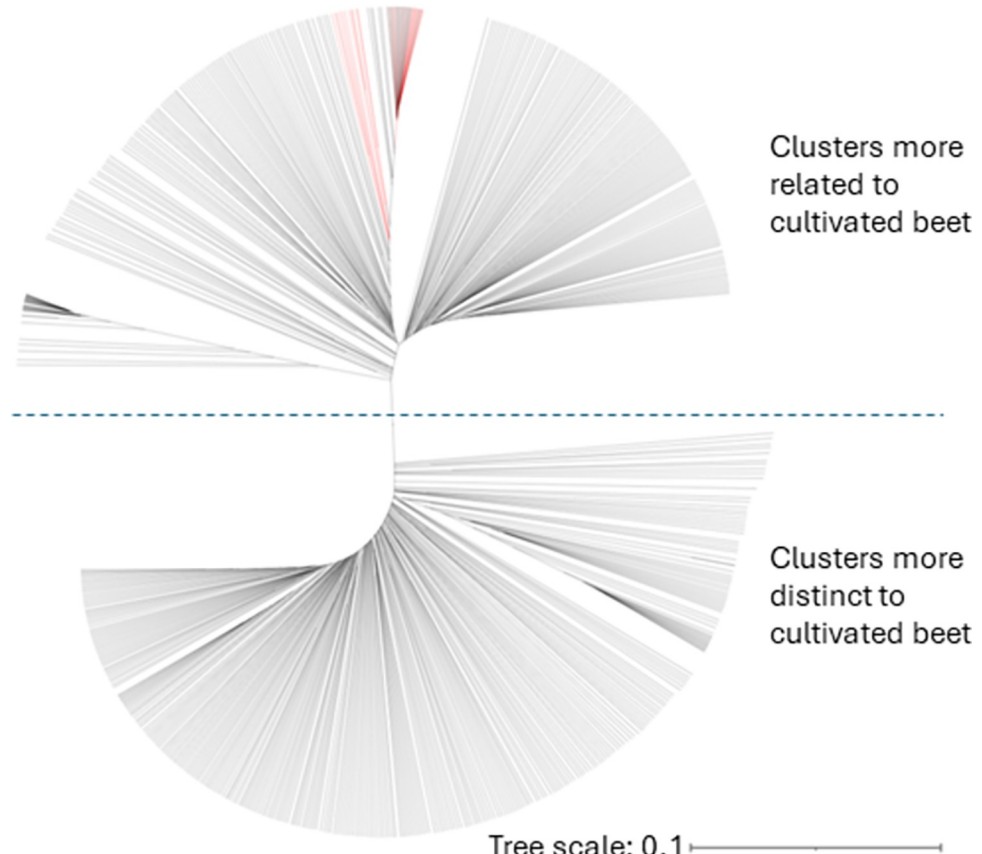

**Fig 2. Phylogenetic tree developed from 599 *Beta maritima* accessions and 30 cultivated beet lines using SNPs covering the whole genome.** The red lines in the cluster clades indicate the cultivated beet lines.

## Population structure in *B. maritima* population

Population structure analysis of *B. maritima* accessions via STRUCTURE indicated that 3, 5, 7, or 8 subpopulations may occur in the collection although 5 or 8 subpopulations was most

**Table 2. Distribution of *Beta maritima* accessions in two distinct phylogenetic clusters defined by phylogenetic analysis using *B. maritima* collection and 30 cultivated beet lines.**

| Cluster | Region | Number of lines | Percentage within cluster (%) |
|---|---|---|---|
| cultivated-related (276 accessions) | | | |
| | Africa | 28 | 10.1 |
| | Asia | 6 | 2.2 |
| | North America | 18 | 6.5 |
| | Northern Europe | 16 | 5.8 |
| | Southern Europe | 168 | 60.9 |
| | Western Europe | 40 | 14.5 |
| cultivated-distinct (323 accessions) | | | |
| | Africa | 30 | 9.3 |
| | North America | 4 | 1.2 |
| | Northern Europe | 165 | 51.1 |
| | Southern Europe | 11 | 3.4 |
| | Western Europe | 113 | 35.0 |

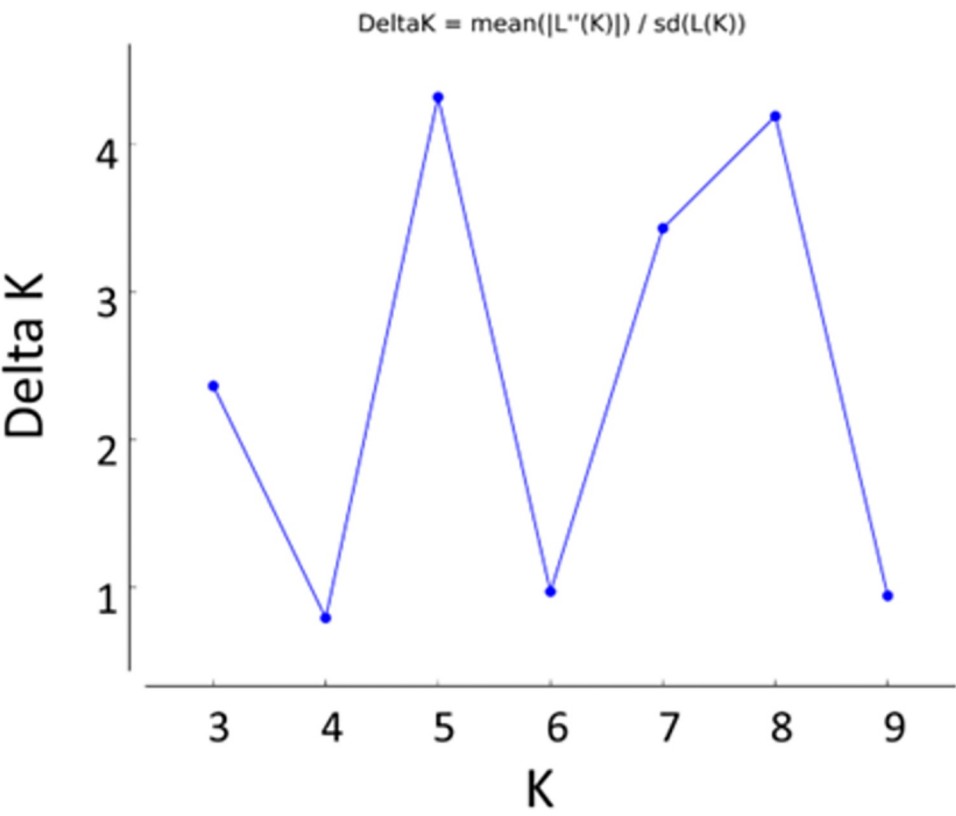

**Fig 3. The plot of $K$ versus $\Delta K$ for determination of the optimum number of subpopulations within 599 *B. maritima* accessions.** Optimal subpopulations are equal to $K$ values with maximal $\Delta K$ values. Data was generated using the computer program STRUCTURE v2.3.4.

likely (Figs 3 and S1). Though no clear population structure could be concluded from STRUCTURE analysis, the grouping of subpopulations by this program agreed with the geographic distribution of the accessions. For example, if the collection was divided into three subpopulations, accessions can be grouped into Atlantic, north African and Mediterranean regions. If five subpopulations were used to describe the collection, subpopulations would group accessions into English Channel, French Atlantic, Moroccan coast near Gibraltar, the Mediterranean Sea, and Levantine and Aegean Sea coasts.

The DAPC method [41] determined that eight clusters was the optimum number of subpopulations in the collection (Figs 4 and S2, S1 Table), which agreed with the cluster grouping in STRUCTURE when $K$ = 8. In detail, Cluster 1 contained 173 accessions with 125 from northern Europe and 46 from western Europe. Cluster 2 had 110 accessions with 59 from southern Europe and 21 from western Europe, but it also contained 22 accessions from north America, 4 accessions from Asia, and 7 accessions from Africa. Cluster 3 only contained 24 accessions with 15 from Morocco. Cluster 4 had 104 accessions with 88 from southern Europe. Cluster 5 carried 114 accessions with 42 from northern Europe (mainly admixture from United Kingdom) and 68 from western Europe (mainly the French Atlantic coast). Cluster 6 included 25 accessions from different regions in addition to the 30 cultivated beets, and thus is corresponding to cultivated beet. Cluster 7 contained 34 accessions with 19 from Egypt and 15 from south of the Aegean Sea in southern Europe, and Cluster 8 had 15 accessions that were all collected from the Atlantic coast of Morrocco (Tables 3 and S1; Fig 4). Overall, accessions in Clusters 2, 3 and 4 are closest to the cultivated cluster (Cluster 6), those in Cluster 7 are the

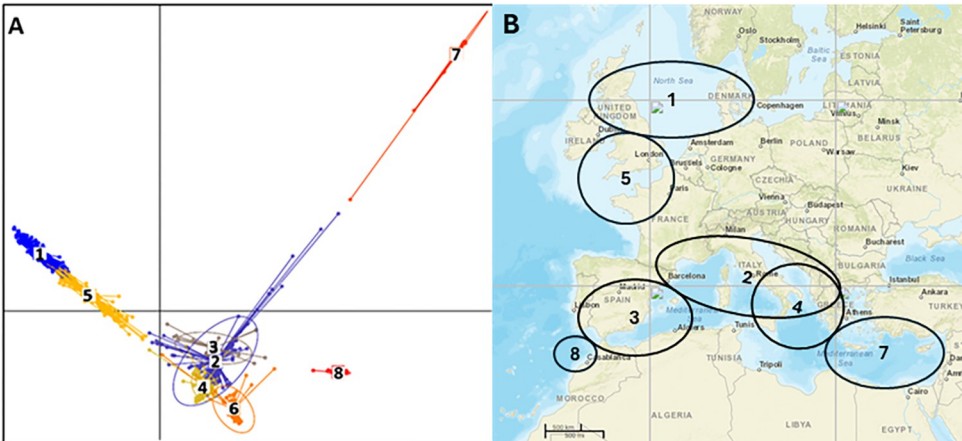

**Fig 4. Genetic structure of the 599 *B. maritima* accessions and 30 cultivated beet lines analyzed using DAPC (discriminant analysis of principal components) methodology.** A. Genetic relationships amongst clusters. The length of bars for each cluster indicates variations within each cluster. Bars in different color correspond to different clusters. B. Geographic distribution of clusters defined by DAPC method. The geographic map is based on NASA world map (https://data.nasa.gov/). Detailed list of accessions in each cluster is available in Tables 3 and S1. Location of Cluster 6 is not shown since it is mainly corresponding to worldwide collected cultivated beet lines.

most distinct from cultivated beet lines. Clusters 1, 5 and 7 are all distinct to the cultivated cluster but accessions in Clusters 1 and 5 showed closer relationship which agrees that two clusters were physically close along the north Atlantic Ocean seashores. Interestingly, the geographic distribution of clusters agreed well with the marine current direction in Mediterranean Sea [51] and the north Atlantic Ocean [52] (S3 Fig), suggesting genetic divergence in *B. maritima* likely caused by the marine currents, facilitating the dispersion *B. maritima* to different regions and subsequent isolation.

## Genetic differentiation among *B. maritima* subpopulations

Pairwise comparison of the fixation index Fst among all subpopulations revealed moderate to high population differentiation (Table 4), which agreed with results from the DAPC analysis above. Low comparison Fst values (high genetic similarity) were found between Clusters 2 and

**Table 3. Distribution of *Beta maritima* accessions in eight clusters as defined by DAPC (discriminant analysis of principal components) methodology across all collection regions[a].**

| Cluster | Cultivated beet | Africa (Egypt) | Africa (Morocco) | Asia | North America | North Europe | South Europe | West Europe | Total |
|---|---|---|---|---|---|---|---|---|---|
| 1 | 0 | 0 | 0 | 0 | 1 | **125** | 1 | **46** | 173 |
| 2 | 0 | 7 | 0 | 4 | 16 | 3 | **59** | **21** | 110 |
| 3 | 0 | 0 | **15** | 0 | 2 | 0 | 7 | 0 | 24 |
| 4 | 0 | 0 | 1* | 0 | 1 | 0 | **88** | 14 | 104 |
| 5 | 0 | 0 | 0 | 0 | 2 | 42 | 2 | **68** | 114 |
| 6 | **30** | 0 | 1 | 2 | 0 | 11 | 7 | 4 | 55 |
| 7 | 0 | **19** | 0 | 0 | 0 | 0 | 15 | 0 | 34 |
| 8 | 0 | 0 | **15** | 0 | 0 | 0 | 0 | 0 | 15 |
| **Total** | 30 | 26 | 32 | 6 | 22 | 181 | 179 | 153 | 629 |

[a] The numbers in bold font style indicate majority of accessions in the corresponding cluster.

* This is the only accession from Tunisia.

**Table 4. Genetic diversity among subpopulations represented by fixation index (Fst) and Nei's genetic distance estimated using a worldwide collection of 599 *Beta maritima* accessions with 30 cultivated beet lines [a].**

| Cluster | 1 | 2 | 3 | 4 | 5 | 6 | 7 | 8 |
|---|---|---|---|---|---|---|---|---|
| 1 | - | 0.31 | 0.20 | 0.31 | 0.05 | 0.47 | 0.53 | 0.33 |
| 2 | 0.77 | - | 0.19 | 0.06 | 0.16 | 0.17 | 0.37 | 0.34 |
| 3 | 0.32 | 0.41 | - | 0.16 | 0.11 | 0.46 | 0.57 | 0.27 |
| 4 | 0.69 | 0.10 | 0.30 | - | 0.17 | 0.28 | 0.44 | 0.34 |
| 5 | 0.06 | 0.38 | 0.22 | 0.38 | - | 0.33 | 0.44 | 0.26 |
| 6 | 0.82 | 0.19 | 0.80 | 0.39 | 0.64 | - | 0.65 | 0.62 |
| 7 | 0.80 | 0.56 | 0.82 | 0.70 | 0.92 | 0.83 | - | 0.75 |
| 8 | 0.49 | 0.73 | 0.30 | 0.61 | 0.45 | 0.77 | 0.95 | - |

[a] Table entries above the diagonal show the calculated Weir and Cockerham's Fst; entries below the diagonal show estimates of Nei's genetic distance.

4, and between Clusters 1 and 5, whereas high Fst values (low genetic similarity) were observed when comparing Clusters 7 or 8 with any other clusters, which agreed with the geographic origins of Clusters 7 and 8 that are more isolated. Clusters 1 and 5 were also distinct from the others as their Fst values, when comparing with other clusters, were relatively high (Table 4), fitting with their locations at north Atlantic Ocean seashores that are far away from other locations. In contrast, Fst values of population comparisons among Clusters 2, 3 and 4 from the seashores of the Mediterranean Sea including the Balearic and Alboran seas were relatively low, which agrees with their close geographic distances or the circular marine current prevailing in those waters [51]. Nei's genetic distance estimated from comparison among clusters agreed well with the corresponding Fst values observed (Table 4).

Comparison of major allele frequencies along chromosomes across all eight clusters found all chromosomes carried regions that had higher levels of variation for allele distribution (Fig 5 and S2 Table), of which, chromosomes 1, 7 and 9 had relatively larger regions with higher allelic variation, and chromosomes 3, 5 and 6 had relatively lower variation. Regions with higher allelic variation may carry genes that help sea beet adapt to different conditions and result in divergence of *B. maritima* during evolution. In addition, major allele frequency in cultivated beet cluster (cluster 6) remained higher than the frequencies in other clusters (Fig 5), consistent with the fact that cultivated beet has undergone higher selection pressure.

### Genetic diversity within *B. maritima* subpopulations

The scaled nucleotide diversity (π), Watterson estimator (ø) and Tajima's D value indicate genetic diversity in subpopulations with less interference due to subpopulation size. Estimation of these scaled genetic diversity indices in eight subpopulation of *B. maritima* defined by the DAPC method indicated that Clusters 1 through 5 had similar diversity with no obvious selection pressure (Table 5). However, Cluster 5, that contains accessions mainly from the French Atlantic coast and seashore of the United Kingdom, exhibited slightly higher but non-significant level of diversity. Clusters 2, 3 and 4 showed a little higher allelic richness when combining the estimations of nucleotide diversity and Watterson estimator, in agreement with the circular marine current in the western Mediterranean Sea and the Tyrrhenian Sea that likely increased allele exchange among these three clusters (Table 5 and S3 Fig).

Clusters 6, 7 and 8 showed a relatively low level of genetic diversity with the estimated Tajima's D being negative (Table 5), indicating selection pressure had been applied to those clusters. Cluster 6 contains mainly cultivated beet lines and agrees with beet cultivars have been under through high selection pressure. The non-natural gene frequency change in Clusters 7

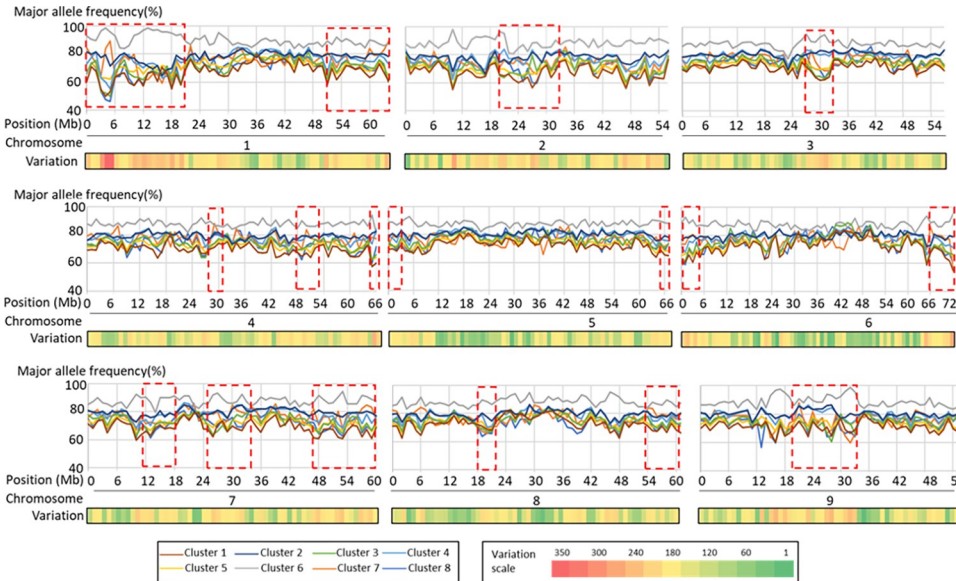

**Fig 5. Major allele frequency distribution on each chromosome across all clusters defined by DAPC (discriminant analysis of principal components) methodology.** Red rectangular boxes with dashed lines indicate genomic regions with a higher level of variation among clusters, and the regions were determined by the above 75th percentile of the major allele frequency variance.

(accessions from north Egypt and south Greece along the Levantine and Aegean Seas) and Cluster 8 (accessions from Morrocco) may be largely due to the smaller population size within clusters, in addition to the possible selection pressure exerted by the unique environmental conditions in these two isolated regions.

## Discussion

The genetic structure of plant populations is influenced by a combination of environment and demographic processes, encompassing historical and contemporary gene flow. Typically, this pattern is assessed indirectly on a broad scale, primarily at the population level [32]. Exploring the mechanisms driving range shifts and factors influencing population formation are crucial questions in evolutionary biology and ecological genetics, particularly within the framework of climate change [19]. The observed patterns in the current study of *B. maritima* populations offer insight into significance of geographic features, contributing to our understanding of

**Table 5. Population genetic indices estimated within each cluster (subpopulation) as defined by DAPC (discriminant analysis of principal components) methodology in 599 *Beta maritima* accessions and 30 cultivated beet lines.**

| Cluster | Scaled ($\pi$) | Watterson scaled | Tajima's D scaled |
|---------|----------------|------------------|-------------------|
| 1 | 0.24 | 0.15 | 0.09 |
| 2 | 0.29 | 0.17 | 0.13 |
| 3 | 0.24 | 0.20 | 0.04 |
| 4 | 0.26 | 0.17 | 0.10 |
| 5 | 0.30 | 0.17 | 0.14 |
| 6 | 0.15 | 0.16 | -0.01 |
| 7 | 0.07 | 0.10 | -0.03 |
| 8 | 0.12 | 0.13 | -0.01 |

how these factors jointly influence the distribution of genetic diversity and population structure at the broader levels of whole population, as well as at individual genetic levels amongst the accessions in the subpopulations.

Sandell et al. [33] found the *B. maritima* accessions they investigated could be divided into two genetically distinct groups. In this study, our phylogenetic analysis of 599 *B. maritima* accessions and 30 cultivated beet lines also revealed two major groups with one being closely related and the other distinct to cultivated beet lines. The cultivated-related group had accessions from the Mediterranean Sea coast whereas the cultivated-distinct group contained accessions predominantly from the seashores of the north Atlantic Ocean. Distinctions between Atlantic and Mediterranean populations of *B. maritima* were also documented in other reports [32,53], and this phylogeographic distribution pattern has also been indicated by variations in chloroplast [13] and patterns of allozyme [54].

By using DAPC analysis, eight distinct clusters of *B. maritima* were identified and each clearly aligned to a defined geographic region. The DAPC analysis in this research largely agrees with findings of Andrello et al. [32] that grouped 1080 accessions of *Beta* germplasm lines into nine clusters. Our analysis focused mainly on *B. maritima* accessions and enabled us to assign each cluster to a specific geographical region. All clusters from DAPC analysis represent segments of a continuous genetic gradient extending from south-east to north-west, which indicates that the genetic structure is influenced by a combination of local genetic drift and limited gene flow.

In *B. maritima*, gene flow can occur through pollen dispersal and the spread of seed. Pollen flow is constrained by factors such as wind and plant density and occurs primarily over short distances [55]. Seed dispersal, on the other hand, is facilitated by seawater movements, leading to seeds being washed away from the shores [13,31]. Combining our *B. maritima* clusters defined by DAPC with marine current direction data in the Mediterranean Sea [51] and Atlantic Ocean [52], the observed patterns of population structure align well with the marine circulation, as indicated in previous reports [19,56]. Our findings suggest a comprehensive understanding of the genetic architecture in *B. maritima*, where both spatial and genetic factors play integral roles.

The Mediterranean region has been recognized as the diversity center of *B. maritima* [10,13,53,56]. Of the eight clusters identified in this research, Cluster 2 contained the most accessions from the coast of western and southern Europe (France, Italy and Greece) along the Mediterranean Sea, and it also contains accessions collected from almost all regions including Asia, Africa, North America, and other regions of Europe (S1 Table), which indicated that region of Cluster 2 is likely to be the center of origin of *B. maritima*. Cluster 4, with a majority of accessions from southern Europe (Italy and Greece), exhibited the closest genetic distance to Cluster 2 (Table 4) and cultivated beet, i.e., Cluster 6 (Fig 4), suggesting it is likely the center of origin for cultivated beet domestication. Sandell et al. [33] further suggested that Greece might have served as the site of domestication for the ancestral sugarbeet, as *B. maritima* accessions in that region were found to be the closest wild relatives to sugarbeet.

Clusters 1, 3, 5, 7 and 8 are likely the result of colonization of seeds spread from the Mediterranean region. Cluster 7 contains accessions from Egypt and southern Greece, and a few from Turkey and southern Italy. Cluster 7 likely originated from sea beet drift from southern Europe and followed by colonization and spread along the shores of the Levantine Sea through the circular marine current of that region (S3 Fig). Cluster 3 could be accessions that originated from Cluster 2 and colonized the southwest seashore of the western Mediterranean (S3 Fig). Due to the circular marine currents within the western Mediterranean and Tyrrhenian Seas, accessions in Cluster 3 were most closely related to accessions in Clusters 2 and 4 (Fig 3). Accessions in Cluster 8 may be originated from Cluster 3 and colonized the Atlantic seashore

of Morrocco since accessions in Cluster 8 were closely related to Cluster 3 (Table 4). Clusters 1 and 5 may also have originated from Cluster 3 but were dispersed via the marine currents of the Atlantic Ocean towards the north. This observation is supported by the low comparison values of Fst and Nei's distance between Cluster 3 and either Cluster 1 or 5, which suggest a relatively close relationship among these clusters (Table 4). Comparison of DNA sequences of genomic regions exhibiting higher allele frequency variation (Fig 5) may facilitate the identification of genes to help elucidate the evolution of *B. maritima*.

Genetic diversity within each subpopulation provided additional insight supporting the potential divergence process of *B. maritima*. The higher allelic richness observed in clusters 2, 3, and 4 aligns with increased chances for allele exchange amongst them due to circular marine currents across the western Mediterranean and Tyrrhenian Sea (S3 Fig). Accessions within these clusters would constitute valuable genetic resources for identifying novel alleles. The low level of genetic diversity within accessions in Clusters 7 and 8 is consistent with the geographic isolation of these accessions due to marine current direction, which limited their allele exchange with accessions in other clusters, consequently reducing allelic richness (Fig 5). The narrow genetic basis due to limited allele richness in two clusters further increased their vulnerability to environmental conditions further reducing allelic richness by environmental selection pressure (Table 5). Our finding agreed with Veloso et al. [57] for role of ocean currents in species dispersal. Overall, accessions in clusters 1, 5, 7 and 8 that are more distinct from clusters 2, 3, 4, and 6 are likely to provide an important source of novel variations to broaden the genetic base and increase the sustainability of sugarbeet, and they need to be considered for prior conservation to maintain genetic diversity.

In summary, an analysis of 599 *B. maritima* accessions and 30 cultivated beet lines using DAPC methodology grouped the publicly available accessions of *B. maritima* into eight subpopulations, with each subpopulation localizing to a distinct geographic region. Population structure analysis elucidated the clinal variation and admixture patterns during evolution in the *B. maritima* collection. Further, the genetic diversity analysis among subpopulations confirmed southern Europe's role as the center of diversity for *B. maritima*, identified relationships among subpopulations, and revealed patterns of dynamic gene flow contributing to *B. maritima* divergence. The knowledge obtained from this study offers new insights that will be valuable for the preservation of genetic diversity within B. *maritima* and the effective utilization of novel alleles and variations to enhance sugar beet improvement.

## Supporting information

**S1 Fig. Population structure of 599 *B. maritima* accessions as a function of the number of subpopulations (*K*) between 2 to 10.** Different colors in each row represent different subpopulations. Each bar represents the estimated membership of subpopulations for a single genotype.
(TIF)

**S2 Fig. Average Bayesian Information Criterion (BIC) plotted against 1–20 putative clusters to provide the optimum number of subpopulations, equivalent to the number at the lowest position, in the world-wide collection of 599 *B. maritima* accessions.**
(TIF)

**S3 Fig.** Geographical distribution of *B. maritima* clusters along with the marine current direction in the Atlantic Ocean (A) and the Mediterranean Sea (B). The geographic map is based on NASA world map (https://data.nasa.gov/). Black arrows indicate marine current direction of the Atlantic Ocean according to Offshore Engineering [52] and that of the Mediterranean Sea

according to Pascual et al. [51]. Red circles with numbers indicate the clusters defined by DAPC (discriminant analysis of principal components) methodology with Cluster 1 contained accessions from northern and western Europe, Cluster 2 had accessions mainly from southern and western Europe, and few accessions from north America, Asia and Africa, Cluster 3 contained accessions from Morocco, Cluster 4 had accessions from southern Europe, Cluster 5 carried accessions from northern Europe (mainly from United Kingdom) and western Europe (mainly from Atlantic coast of French), Cluster 6 is corresponding to 30 cultivated beets used as the reference (it's not shown in the picture), Cluster 7 contained accessions from Egypt and south of the Aegean Sea in southern Europe, and Cluster 8 had accessions collected from the Atlantic coast of Morrocco.
(TIF)

**S1 Table. List of 599 *B. maritima* accessions and 30 cultivated beet lines in eight clusters as defined by DAPC (discriminant analysis of principal components) methodology with collection locations for each accession.**
(DOCX)

**S2 Table. Average major allele frequencies on each chromosome of *B. maritima* across the eight clusters as defined by DAPC (discriminant analysis of principal components) methodology.**
(DOCX)

## Acknowledgments

This research is supported by the USDA-ARS CRIS project No. 3060-21000-045-000D, the Beet Sugar Development Foundation (BSDF), and the Sugarbeet Research and Education Board of Minnesota and North Dakota (SBREB). Mention of trade names or commercial products in this article is solely for the purpose of providing specific information and does not imply recommendation or endorsement by the U.S. Department of Agriculture. The U.S. Department of Agriculture is an equal opportunity provider and employer.

## Author Contributions

**Conceptualization:** Chenggen Chu.

**Data curation:** Chenggen Chu.

**Formal analysis:** Muhammad Massub Tehseen, Nathan A. Wyatt, Karen K. Fugate, Lisa S. Preister, Shengming Yang, Vanitharani Ramachandran, Xuehui Li, Chenggen Chu.

**Investigation:** Chenggen Chu.

**Methodology:** Muhammad Massub Tehseen, Karen K. Fugate, Lisa S. Preister, Shengming Yang, Vanitharani Ramachandran, Xuehui Li, Chenggen Chu.

**Project administration:** Melvin D. Bolton.

**Resources:** Melvin D. Bolton.

**Supervision:** Melvin D. Bolton.

**Writing – original draft:** Muhammad Massub Tehseen, Chenggen Chu.

**Writing – review & editing:** Nathan A. Wyatt, Melvin D. Bolton, Karen K. Fugate, Lisa S. Preister, Shengming Yang, Vanitharani Ramachandran, Xuehui Li, Chenggen Chu.

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
