## [Decision Letter · Decision Letter 0]

4 Jun 2024

PONE-D-24-16245Genetic drift, historic migration, and limited gene flow contributing to the subpopulation divergence in wild sea beet (Beta vulgaris ssp. maritima)PLOS ONE

Dear Dr. chu,

Thank you for submitting your manuscript to PLOS ONE. After careful consideration, we feel that it has merit but does not fully meet PLOS ONE’s publication criteria as it currently stands. Therefore, we invite you to submit a revised version of the manuscript that addresses the points raised during the review process.

We look forward to receiving your revised manuscript.

Kind regards,

Awatif Abid Al-Judaibi, PhD

Academic Editor

PLOS ONE

Journal Requirements:

"USDA-ARS CRIS project No. 3060-21000-045-000D, the Beet Sugar Development Foundation (BSDF), and the Sugarbeet Research and Education Board of Minnesota and North Dakota (SBREB)."

4. In the online submission form, you indicated that [All relevant data are within the manuscript and its Supporting Information files. Raw data are available per request.]. 

5. We note that [Figure 4 and S3] in your submission contain [map/satellite] images which may be copyrighted. All PLOS content is published under the Creative Commons Attribution License (CC BY 4.0), which means that the manuscript, images, and Supporting Information files will be freely available online, and any third party is permitted to access, download, copy, distribute, and use these materials in any way, even commercially, with proper attribution. For these reasons, we cannot publish previously copyrighted maps or satellite images created using proprietary data, such as Google software (Google Maps, Street View, and Earth). For more information, see our copyright guidelines: http://journals.plos.org/plosone/s/licenses-and-copyright.

a. You may seek permission from the original copyright holder of Figure 4 and S3 to publish the content specifically under the CC BY 4.0 license.  

Reviewers' comments:

Reviewer's Responses to Questions

**Comments to the Author**

1. Is the manuscript technically sound, and do the data support the conclusions?

Reviewer #1: Yes

Reviewer #2: Yes

Reviewer #3: Yes

2. Has the statistical analysis been performed appropriately and rigorously? 

Reviewer #1: Yes

Reviewer #2: Yes

Reviewer #3: I Don't Know

3. Have the authors made all data underlying the findings in their manuscript fully available?

Reviewer #1: Yes

Reviewer #2: Yes

Reviewer #3: Yes

4. Is the manuscript presented in an intelligible fashion and written in standard English?

Reviewer #1: Yes

Reviewer #2: Yes

Reviewer #3: Yes

5. Review Comments to the Author

Reviewer #1: First, I extend my gratitude to the authors for their outstanding work and well-constructed text. In this study, authors investigated 599 sea beet (Beta maritima) accessions collected from the north Atlantic Ocean and Mediterranean Sea coasts. It is a comprehensive study about the distribution and origin of sea beet and cultivated beet. The title explains the research properly, and the results are described well in details. I think the manuscript contains a good research question and explains the migration route of wild beet along the north Atlantic Ocean and Mediterranean Sea coasts. In the future, the findings will be useful to identify novel genomic regions related to resistance against abiotic/biotic stress factors in Beta vulgaris. However, I have some comments as follows. I recommend minor revisions.

Here are my comments:

-In Materials and Methods, authors did not describe the growth conditions of beet accessions. Authors should mention how many accessions they used from each of the following: Africa, Asia, northern Europe, southern Europe, western Europe, eastern Europe, and North America.

-In Figures 4 and S3, why did not authors show the location of Cluster 6? That map does not show the locations correctly unless I have missed something. For example, I do not see Greece and Italy in Cluster 2 (please see discussion, line 306). Why do you think the Cluster 2 is the center of origin for sea beet and cultivated beet? You may highlight the experimental results for clearer explanation. In addition, each cluster (1-8) must be briefly described in the legend of figure S3.

Reviewer #2: Correct name: Beta vulgaris subsp. maritima (L.) Arcang.

Adding to the title and in the first citation of the text the authorship of the species -> Verify: www.tropicos.org

I suggest adding the botanical family name to the title or in the keywords.

I suggest adding the scientific name of the causal agents of the diseases mentioned in paragraph 3 of the Introduction.

I congratulate the authors on this present work, of fundamental importance for the establishment or better management of genetic improvement programs, as well as for the conservation of wild species.

Reviewer #3: The origin of cultivated plants and the identification of the most closely related native populations, as well as their crop wild relatives, is a highly topical subject, given its potential for the genetic improvement of crops. Following on from other work that has been carried out on sugar beet, the submitted manuscript analyses more than six hundred samples of Beta maritima from various regions of the world. The results presented are interesting and improve the knowledge of the phylogenetic relationships of the populations, particularly those occurring in the Mediterranean region.

In general, the manuscript is well-written, well-structured, and easy to understand.

Apart from a few minor suggestions/comments (listed below), there is one aspect that I would like to highlight because I think it should be corrected: some of the samples used are from areas where Beta maritima is not native, but naturalised (this is the case with the samples from North America). I think this should be mentioned/analysed throughout the text (material and methods, results and discussion, tables, etc.). This aspect can even add value to the discussion, as hypotheses can be put forward for the origin of the populations in the regions where Beta maritima is naturalised.

Other comments:

Ln. 54 – ‘Sea beet (Beta vulgaris L. ssp. maritima Arcang., referred to as B. maritima hereafter)’ – Currently, some of the world's most important taxonomic databases (e.g. Plants of the World Online - POWO; and World Flora Online - WFO) only accept Beta vulgaris subsp. vulgaris and consider Beta vulgaris subsp. maritima as a synonym. This does not invalidate the work developed, but it can be mentioned in the text that the work carried out used the classification adopted by the US National Plant Germplasm System (NPGS).

ln. 59 – ‘Wild B. maritima populations extend along the Mediterranean basin and the Atlantic coast.’ – This description should be more detailed (see POWO) and the regions where the species is introduced should be indicated.

Ln. 76 – ‘Tehseen et al’ - Tehseen et al. The full stop is missing.

Ln. 108 to 111 – Detail the distribution (native or non-native) in the mentioned regions. In Table 1, mark (e.g. with a *) the accessions collected in regions where the taxa is not native (e.g. North America). Check the values in this table: the total is 601 and not 599 as it should be.

Check the legend of Fig. 1 (ln 524) - references to A and B are missing.

In Results section, I think Fig. 3 (Ln. 208) could be made supplementary and Fig. S3 (ln. 231) be included in the main text. The latter is quite enlightening. However, its graphic presentation could be improved.

ln. 273 – ‘… is influenced by a combination of landscape characteristics.’ - I suggest replacing 'landscape' (which has aesthetic connotations) with environment.

Ln. 300 – In the discussion, this manuscript could also consider the results of the paper Veloso et al. 2021 (Genetic Diversity and Population Structure of Wild Beets (Beta spp.) from the Western Iberian Peninsula and the Azores and Madeira Islands) which also highlights the importance of ocean currents in species dispersal.

Finally, I suggest complementing the discussion by including some conservation proposals, namely the prioritisation of certain populations / regions taking into account their phylogenetic affinities and genetic diversity.

6. PLOS authors have the option to publish the peer review history of their article (what does this mean?). If published, this will include your full peer review and any attached files.

Reviewer #1: No

Reviewer #2: No

Reviewer #3: No

---

## [Author Response · Author response to Decision Letter 0]

7 Jun 2024

Response to Editor:

Response: format was checked and corrected according to the guidelines.

Response: grant numbers added.

"USDA-ARS CRIS project No. 3060-21000-045-000D, the Beet Sugar Development Foundation (BSDF), and the Sugarbeet Research and Education Board of Minnesota and North Dakota (SBREB)."

Response: the statement is added.

4. In the online submission form, you indicated that [All relevant data are within the manuscript and its Supporting Information files. Raw data are available per request.]. 

Response: All data are available and attached, and the raw marker data is stored at NPGS-GRIN and is publicly available. This information is added in the manuscript.

5. We note that [Figure 4 and S3] in your submission contain [map/satellite] images which may be copyrighted. All PLOS content is published under the Creative Commons Attribution License (CC BY 4.0), which means that the manuscript, images, and Supporting Information files will be freely available online, and any third party is permitted to access, download, copy, distribute, and use these materials in any way, even commercially, with proper attribution. For these reasons, we cannot publish previously copyrighted maps or satellite images created using proprietary data, such as Google software (Google Maps, Street View, and Earth). For more information, see our copyright guidelines: http://journals.plos.org/plosone/s/licenses-and-copyright. 

a. You may seek permission from the original copyright holder of Figure 4 and S3 to publish the content specifically under the CC BY 4.0 license. 

Response: the figures were reproduced using NASA world map (https://data.nasa.gov/), and ocean current direction was manually added according to the reference listed.

Response: added.

Response: all references were checked.

Responses to Reviewers' comments:

Reviewer #1: 

First, I extend my gratitude to the authors for their outstanding work and well-constructed text. In this study, authors investigated 599 sea beet (Beta maritima) accessions collected from the north Atlantic Ocean and Mediterranean Sea coasts. It is a comprehensive study about the distribution and origin of sea beet and cultivated beet. The title explains the research properly, and the results are described well in details. I think the manuscript contains a good research question and explains the migration route of wild beet along the north Atlantic Ocean and Mediterranean Sea coasts. In the future, the findings will be useful to identify novel genomic regions related to resistance against abiotic/biotic stress factors in Beta vulgaris. However, I have some comments as follows. I recommend minor revisions.

Here are my comments:

- In Materials and Methods, authors did not describe the growth conditions of beet accessions. 

Response: information was added.

- Authors should mention how many accessions they used from each of the following: Africa, Asia, northern Europe, southern Europe, western Europe, eastern Europe, and North America.

Response: number of accessions were added for each region. The region ‘eastern Europe’ is deleted since the two lines have not good marker data and were not included in the analysis.

-In Figures 4 and S3, why did not authors show the location of Cluster 6? 

Response: it has two reasons: 1) the cluster 6 is corresponding to cultivated beet that selected worldwide, not a specific location is assigned; and 2) B. maritima accessions in the cluster 6 are spread similarly in several other clusters with no dominant amount from a specific region, it is impossible to assign a location for those accessions.

- That map does not show the locations correctly unless I have missed something. For example, I do not see Greece and Italy in Cluster 2 (please see discussion, line 306). 

Response: Thank you so much for pointing this out. When making the figure, we only thought to find regions and try to make them distinct, but clusters 2 and 4 are very close and their regions are largely overlapped. We thus enlarged the region of the cluster 2 to let it covers Italy and Greece, and it is more consistent with our results. Great thanks! 

- Why do you think the Cluster 2 is the center of origin for sea beet and cultivated beet? You may highlight the experimental results for clearer explanation. 

Response: Cluster 2 is the center of origin for sea beet simply because it contains accessions collected from almost all regions, the location of regions for Cluster 2 is determined based on the location that majority of the accessions were collected. Accessions in Cluster 2 but collected from the other regions are indication of they were migrated from the region we defined, indicating regions assigned to major accessions in Cluster 2 at Mediterranean Sea coast likely to be the center of sea beet. However, the sea beet accessions close to Cluster 6 that equivalent to cultivated beet were largely collected in Greece, indicating Greece might be the center of cultivated beet domesticated from sea beet.

- In addition, each cluster (1-8) must be briefly described in the legend of figure S3.

Response: Great thanks for this suggestion, cluster descriptions were added accordingly. We appreciate your comments to improve the manuscript.

Reviewer #2: 

- Correct name: Beta vulgaris subsp. maritima (L.) Arcang. Adding to the title and in the first citation of the text the authorship of the species -> Verify: www.tropicos.org I suggest adding the botanical family name to the title or in the keywords.

Response: Great thanks, the scientific name was corrected in the title and in the text.

- I suggest adding the scientific name of the causal agents of the diseases mentioned in paragraph 3 of the Introduction.

Response: the causal agents of the diseases were added.

- I congratulate the authors on this present work, of fundamental importance for the establishment or better management of genetic improvement programs, as well as for the conservation of wild species.

Response: great thanks and appreciate your comments for improving the manuscript.

Reviewer #3: 

The origin of cultivated plants and the identification of the most closely related native populations, as well as their crop wild relatives, is a highly topical subject, given its potential for the genetic improvement of crops. Following on from other work that has been carried out on sugar beet, the submitted manuscript analyses more than six hundred samples of Beta maritima from various regions of the world. The results presented are interesting and improve the knowledge of the phylogenetic relationships of the populations, particularly those occurring in the Mediterranean region.

In general, the manuscript is well-written, well-structured, and easy to understand.

Apart from a few minor suggestions/comments (listed below), there is one aspect that I would like to highlight because I think it should be corrected: some of the samples used are from areas where Beta maritima is not native, but naturalised (this is the case with the samples from North America). I think this should be mentioned/analysed throughout the text (material and methods, results and discussion, tables, etc.). This aspect can even add value to the discussion, as hypotheses can be put forward for the origin of the populations in the regions where Beta maritima is naturalised.

Response: We agree the location of some accessions not the location they were collected, as you indicated that accessions from North America may be donated by some institution oversea. Since the number of such accessions are small and they didn’t affect the results much, we didn’t mention about them in the discussion. Actually, we just started a new project using conserved gene sequence to investigate the evolution of B. maritima accessions as well as cultivated beets, which will be better to differentiate B. maritima accessions and then narrow down region of the origin. 

Other comments:

Ln. 54 – ‘Sea beet (Beta vulgaris L. ssp. maritima Arcang., referred to as B. maritima hereafter)’ – Currently, some of the world's most important taxonomic databases (e.g. Plants of the World Online - POWO; and World Flora Online - WFO) only accept Beta vulgaris subsp. vulgaris and consider Beta vulgaris subsp. maritima as a synonym. This does not invalidate the work developed, but it can be mentioned in the text that the work carried out used the classification adopted by the US National Plant Germplasm System (NPGS).

Response: Thanks. We made the change accordingly.

ln. 59 – ‘Wild B. maritima populations extend along the Mediterranean basin and the Atlantic coast.’ – This description should be more detailed (see POWO) and the regions where the species is introduced should be indicated.

Response: More details were added according to POWO, but we are not sure if it’s correct for mentioning accessions were introduced. I heard that few B. maritima accessions from California, USA might be natively collected, but no further information to indicate if it’s true.

Ln. 76 – ‘Tehseen et al’ - Tehseen et al. The full stop is missing.

Response: Corrected. Thanks!

Ln. 108 to 111 – Detail the distribution (native or non-native) in the mentioned regions. In Table 1, mark (e.g. with a *) the accessions collected in regions where the taxa is not native (e.g. North America). Check the values in this table: the total is 601 and not 599 as it should be.

Response: Detailed information was added, accordingly. It should be 599 accessions. The two accessions from eastern Europe have not good marker data and were removed from analysis later, but we forget to change the table, accordingly.

Check the legend of Fig. 1 (ln 524) - references to A and B are missing.

Response: Thanks for finding this. We changed the figure but forget to change the figure legends. It’s now corrected.

In Results section, I think Fig. 3 (Ln. 208) could be made supplementary and Fig. S3 (ln. 231) be included in the main text. The latter is quite enlightening. However, its graphic presentation could be improved.

Response: Thanks for this comment. We originally put Figure S3 in the main text, but considered the figure was published, we then move it as supplemental file. This did flag out by academic editor and the journal. We have to redo the figure but still think it’s better to put it as the supplementary.

ln. 273 – ‘… is influenced by a combination of landscape characteristics.’ - I suggest replacing 'landscape' (which has aesthetic connotations) with environment.

Response: Thanks. We changed it as suggested.

Ln. 300 – In the discussion, this manuscript could also consider the results of the paper Veloso et al. 2021 (Genetic Diversity and Population Structure of Wild Beets (Beta spp.) from the Western Iberian Peninsula and the Azores and Madeira Islands) which also highlights the importance of ocean currents in species dispersal.

Response: Thanks. The reference is cited.

Finally, I suggest complementing the discussion by including some conservation proposals, namely the prioritisation of certain populat

---

## [Decision Letter · Decision Letter 1]

29 Jul 2024

Genetic drift, historic migration, and limited gene flow contributing to the subpopulation divergence in wild sea beet (Beta vulgaris ssp. maritima (L.) Arcang)

PONE-D-24-16245R1

Dear Dr. Chenggen Chu,

We’re pleased to inform you that your manuscript has been judged scientifically suitable for publication and will be formally accepted for publication once it meets all outstanding technical requirements.

Kind regards,

Awatif Abid Al-Judaibi, PhD

Academic Editor

PLOS ONE

Reviewers' comments:

Reviewer's Responses to Questions

**Comments to the Author**

1. If the authors have adequately addressed your comments raised in a previous round of review and you feel that this manuscript is now acceptable for publication, you may indicate that here to bypass the “Comments to the Author” section, enter your conflict of interest statement in the “Confidential to Editor” section, and submit your "Accept" recommendation.

Reviewer #1: (No Response)

Reviewer #3: All comments have been addressed

2. Is the manuscript technically sound, and do the data support the conclusions?

Reviewer #1: Yes

Reviewer #3: (No Response)

3. Has the statistical analysis been performed appropriately and rigorously? 

Reviewer #1: Yes

Reviewer #3: (No Response)

4. Have the authors made all data underlying the findings in their manuscript fully available?

Reviewer #1: Yes

Reviewer #3: (No Response)

5. Is the manuscript presented in an intelligible fashion and written in standard English?

Reviewer #1: Yes

Reviewer #3: (No Response)

6. Review Comments to the Author

Reviewer #1: The authors have improved the manuscript based on the reviewers' comments, I congratulate them for this comprehensive study. I recommend acceptance.

Reviewer #3: (No Response)

7. PLOS authors have the option to publish the peer review history of their article (what does this mean?). If published, this will include your full peer review and any attached files.

Reviewer #1: **Yes: **Seher Yolcu

Reviewer #3: No

---

## [Editor Report · Acceptance letter]

27 Aug 2024

PONE-D-24-16245R1 

PLOS ONE

Dear Dr. Chu, 

I'm pleased to inform you that your manuscript has been deemed suitable for publication in PLOS ONE. Congratulations! Your manuscript is now being handed over to our production team.

Kind regards, 

on behalf of

Professor Awatif Abid Al-Judaibi 

Academic Editor

PLOS ONE